Estimating the impact of influenza on the epidemiological dynamics of SARS-CoV-2

Domenech de Cellès Matthieu domenech@mpiib-berlin.mpg.de 1
Casalegno Jean-Sebastien 2 3
Lina Bruno 2 3
Opatowski Lulla 4 5
1 Infectious Disease Epidemiology Group, Max Planck Institute for Infection Biology , Berlin , Germany
2 Laboratoire de Virologie des HCL, IAI, CNR des Virus à Transmission Respiratoire (dont la grippe) Hôpital de la Croix-Rousse F-69317 Lyon Cedex 04, France , Lyon , France
3 Virpath, Centre International de Recherche en Infectiologie (CIRI), Université de Lyon Inserm U1111, CNRS UMR 5308, ENS de Lyon, UCBL F-69372 , Lyon , France
4 Université Paris-Saclay, UVSQ, Univ. Paris-Sud, Inserm, CESP, Anti-Infective Evasion and Pharma- Coepidemiology Team , Montigny-Le-Bretonneux , France
5 Institut Pasteur, Epidemiology and Modelling of Evasion to Antibiotics , Paris , France
Palazón-Bru Antonio
Electronic publication date: 2021 Dec 3
Publication date: 2021
Volume: 9
Electronic Location ID: e12566
Received 2021 Jun 4; Accepted 2021 Nov 8
Copyright: ©2021 Domenech de Cellès et al.
Copyright year: 2021
Copyright holder: Domenech de Cellès et al.
License: This is an open access article distributed under the terms of the Creative Commons Attribution License, which permits unrestricted use, distribution, reproduction and adaptation in any medium and for any purpose provided that it is properly attributed. For attribution, the original author(s), title, publication source (PeerJ) and either DOI or URL of the article must be cited.
License URL: https://creativecommons.org/licenses/by/4.0/

Keywords: SARS-CoV-2, COVID-19, Influenza, Virus–virus interaction, Mathematical modeling

Funding: No specific funding was used for this study.

==============================
As in past pandemics, co-circulating pathogens may play a role in the epidemiology of coronavirus disease 2019 (COVID-19), caused by the novel severe acute respiratory syndrome coronavirus 2 (SARS-CoV-2). In particular, experimental evidence indicates that influenza infection can up-regulate the expression of ACE2—the receptor of SARS-CoV-2 in human cells—and facilitate SARS-CoV-2 infection. Here we hypothesized that influenza impacted the epidemiology of SARS-CoV-2 during the early 2020 epidemic of COVID-19 in Europe. To test this hypothesis, we developed a population-based model of SARS-CoV-2 transmission and of COVID-19 mortality, which simultaneously incorporated the impact of non-pharmaceutical control measures and of influenza on the epidemiological dynamics of SARS-CoV-2. Using statistical inference methods based on iterated filtering, we confronted this model with mortality incidence data in four European countries (Belgium, Italy, Norway, and Spain) to systematically test a range of assumptions about the impact of influenza. We found consistent evidence for a 1.8–3.4-fold (uncertainty range across countries: 1.1 to 5.0) average population-level increase in SARS-CoV-2 transmission associated with influenza during the period of co-circulation. These estimates remained robust to a variety of alternative assumptions regarding the epidemiological traits of SARS-CoV-2 and the modeled impact of control measures. Although further confirmatory evidence is required, our results suggest that influenza could facilitate the spread and hamper effective control of SARS-CoV-2. More generally, they highlight the possible role of co-circulating pathogens in the epidemiology of COVID-19.

Introduction

The current pandemic of coronavirus disease 2019 (COVID-19), caused by the novel severe acute respiratory syndrome coronavirus 2 (SARS-CoV-2), has led to global alarm. Following the first case reports in December 2019 in Wuhan, China (Zhu et al., 2020), SARS-CoV-2 rapidly spread across the globe and has resulted in approximately 171 million cases and 3.6 million deaths worldwide, as of June 1, 2021 (Johns Hopkins University Center for Systems Science and Engineering (JHU CCSE), 2020). Because of the initial lack of prophylactic or therapeutic treatments, the pandemic caused the implementation of unprecedented control measures, which culminated in the lockdown of several billion people in over 100 countries during April–May 2020 (Hale et al., 2020). Although a number of fixed (e.g., greater age, male sex) and chronic (e.g., hypertension, diabetes) risk factors of mortality have now been identified (Williamson et al., 2020), the time-varying drivers of COVID-19 epidemiology remain poorly understood. Experience gained from past pandemics has highlighted the potentially large contribution of co-circulating pathogens to the burden of an emerging disease (Morens, Taubenberger & Fauci, 2008). Despite the relevance for epidemic forecasting and for designing control strategies, the impact of co-circulating pathogens on SARS-CoV-2 epidemiology has remained largely unexplored (Thindwa et al., 2020).

Respiratory viruses—including SARS-CoV-2 and other coronaviruses, rhinoviruses, influenza viruses, etc.—form a large class of viruses that cause seasonal infections of the respiratory tract in humans. Mounting evidence indicates that their epidemiologies are not independent, as a result of interaction mechanisms that may operate at different scales and that can be classified as either facilitatory or antagonistic (DaPalma et al., 2010; Opatowski, Baguelin & Eggo, 2018). The interaction between the respiratory syncytial virus (RSV) and influenza may provide an example of antagonism. Indeed, experimental evidence in ferrets has shown that influenza viruses induce an antiviral state that transiently limits secondary infection with RSV (Chan et al., 2018), an effect postulated to explain the delayed epidemic of RSV during the 2009 influenza pandemic (Casalegno et al., 2010; Mak et al., 2012). Although such antagonistic interactions appear, to date, to be the most common among respiratory viruses (Opatowski, Baguelin & Eggo, 2018), experimental evidence indicates that co-infections may also have a facilitatory effect, for example by increasing viral growth (Goto et al., 2016). Increased transmission of influenza during co-infection with other respiratory viruses was also proposed to explain the multiple waves during the 1918 influenza pandemic (Merler et al., 2008). Interestingly, according to recent evidence a viral respiratory infection (in particular with influenza viruses) can up-regulate the expression of ACE2—the cognate receptor of SARS-CoV-2 in human cells—in the respiratory epithelium (Smith et al., 2020; Ziegler et al., 2020). In addition, this up-regulation was demonstrated experimentally to increase infectivity of SARS-CoV-2 in mice co-infected with influenza A virus (Bai et al., 2021). This suggests that respiratory viruses could affect the epidemiology of SARS-CoV-2. Here, we hypothesized that influenza—which peaked in February 2020 and therefore co-circulated during the early spread of COVID-19 in Europe (Fig. 1B)—interacted with SARS-CoV-2.

Figure 1 Potential drivers of SARS-CoV-2 transmission in Belgium, Italy, Norway, and Spain.

(A) Time plot of the stringency index, a country-level aggregate measure of the number and of the strictness of non-pharmaceutical control measures implemented by governments. The vertical dashed line indicates the start of the nationwide lockdown (Flaxman et al., 2020). (B) Time plot of influenza incidence, calculated as the product of the incidence of influenza-like illnesses and of the fraction of samples positive to any influenza virus (see also Fig. S1 for a time plot of the latter two variables). The vertical dashed lines delimitate the period of overlap between SARS-CoV-2 and influenza, defined as the period between the assumed start date of SARS-CoV-2 community transmission and 6 weeks after the epidemic peak of influenza (Paget et al., 2020). In each country, the time series displayed were incorporated as covariates, which modulated the transmission rate of SARS-CoV-2 in our model (see Methods). In B, the y-axis values differ for each panel.

To test this hypothesis, we developed a semi-mechanistic, population-based model of SARS-CoV-2 transmission and of COVID-19 mortality. Using likelihood-based statistical inference methods, we fitted this model to mortality incidence data in four European countries to test a range of assumptions about the impact of influenza on the transmission dynamics of SARS-CoV-2. We find that influenza may have transiently increased the transmission of SARS-CoV-2 during the first wave of COVID-19 in Europe. Although further confirmatory evidence is required, our results suggest that influenza could facilitate the spread of SARS-CoV-2 and, more generally, emphasize the potential role of co-circulating pathogens in the epidemiology of COVID-19.

Materials & Methods

Data

Stringency index data

Country-level time series of the stringency index were available from the Oxford COVID-19 Government Response Tracker, developed at the University of Oxford and described elsewhere (Hale et al., 2020). Briefly, the stringency index provides an aggregate measure of the number and of the strictness of non-pharmaceutical control measures implemented by governments in response to the COVID-19 epidemic. The stringency index is defined as the average of nine normalized ordinal variables, which quantify the strength (e.g., recommended or required) and the scope (e.g., targeted or general) of closure and containment measures (8 variables) and of health measures (1 variable). The resulting index allows quantifying the strength of control measures in a systematic way, on a scale ranging from 0 (no interventions) to 100 (maximum number and maximal intensity of control measures). Of note, however, the stringency index does not quantify the impact of control measures, which likely varied across countries (Flaxman et al., 2020). In formulating our model, we therefore modeled the relationship between the stringency index and the relative reduction in SARS-CoV-2 transmission using a non-decreasing function, whose parameters represented the impact of control measures and were estimated from the data.

Influenza incidence data

Virological data on the weekly numbers of samples tested and of samples positive to any influenza virus were available from the FluNet database, compiled by the WHO (Fig. S1A). Parallel syndromic data on the weekly incidence rate of influenza-like illnesses (ILI) were available from the FluID database, also compiled by the WHO (Fig. S1B). These data were deemed high-quality and used in a previous study on influenza forecasting in the countries considered here (Kramer & Shaman, 2019). The weekly incidence rate of influenza was then calculated as the product of ILI incidence and of the fraction of samples positive to any influenza virus (Fig. 1B). Because the magnitude of influenza incidence thus calculated varied markedly across countries (e.g., as a result of different surveillance systems and case definitions), we rescaled each time series by its average during the period of co-circulation of influenza and SARS-CoV-2 (Fig. 1B). The resulting time series was therefore dimensionless and equalled 1 when influenza incidence equalled its average value during that period.

COVID-19 mortality data

Data on the daily number of deaths caused by SARS-CoV-2 (counted by date of death) were available from national public health public institutes, in Belgium (Belgian institute for health, sciensano)(2020), in Italy (Dipartimento della Protezione Civile, 2020), and in Spain (Instituto de Salud Carlos III, official data with historical corrections compiled by the media (DATADISTA, 2020). In Norway, the data were available from the worldwide database compiled by the European Center for Disease Control and Prevention (European Centre for Disease Prevention and Control, 2020). Following a previous study (Flaxman et al., 2020), and to avoid a possible bias caused by the dominance of deaths due to non-locally acquired infections early in the epidemic, we included observed deaths from the date after which the cumulative observed death count exceeded 10. Data points before that date were treated as missing and were assigned a conditional log-likelihood of 0, such that they did not contribute to the overall log-likelihood. The data were not further pre-processed, except in Italy, where a negative death count was reported on 24 June 2020 and was treated as missing and also assigned a log-likelihood of 0.

Transmission model

Model formulation

We formulated a variant of the standard Susceptible–Exposed–Infected–Recovered transmission model (Keeling & Rohani, 2008), using the method of stages to allow for realistic distributions of the latent, infectious, and onset-to-death periods (Lloyd, 2001; Wearing, Rohani & Keeling, 2005). Specifically, we assumed that the latent and infectious periods were Erlang-distributed with shape parameter 2 and mean 1/σ = 4 days and 1/γ = 5 days, respectively (Li et al., 2020). The resulting generation time Tg (i.e., the time from infection of a primary case to transmission to a secondary case) had a mean of 6.5 days and a coefficient of variation of 0.58 (see Fig. S2 for the full distribution and the details of the calculation), to Svensson (2007) and Camacho et al. (2011) consistent with empirical observations and with the values fixed in a previous modeling study (Flaxman et al., 2020; Bi et al., 2020). To model the impact of the gradual implementation of non-pharmaceutical control measures (e.g., border closure, school closure, lockdown), we mapped the stringency index (denoted by si(t)) to the time-varying relative reduction in transmission of SARS-CoV-2 (denoted by rβ(t)). Specifically, we used the following simple linear scaling function, with saturation: rβt= min1,b×sit100

Here the parameter b quantifies how steeply the transmission rate of SARS-CoV-2 decreases as the stringency index increases. Hence, this parameter can be interpreted as a measure of the impact of non-pharmaceutical control measures on SARS-CoV-2 transmission. The deterministic variant of the model was represented by the following set of differential equations: S ˙=−λtSE ˙1=λtS−2σE1E ˙2=2σE1−E2I ˙1=2σE2−2γI1I ˙2=2γI1−I2R ˙=2γI2

The force of infection (that is, the per capita rate at which susceptible individuals contract infection (Keeling & Rohani, 2008)), λ(t), was modeled as: λt=βtI1+I2Nβt=β01−rβtβFtrβt=min1,b×sit100βFt=max0,1+βFFtRet=βtγ×StN

where R0 represents the basic reproduction number of SARS-CoV-2, β0 = R0γ the basic transmission rate, Re(t) the time-varying effective reproduction number, N the population size (assumed constant during the study period), and F(t) the renormalized time series of influenza incidence, incorporated as a covariate into the model (Fig. 1). With this formulation, the parameter βF quantifies the impact of influenza on SARS-CoV-2 transmission: βF > 0 if influenza increases transmission, βF < 0 if influenza decreases transmission, and βF = 0 if influenza has no impact on transmission (null hypothesis). More specifically, the average incidence of influenza during the period of co-circulation with SARS-CoV-2 corresponds to F(t) = 1, such that 1 + βF represents the average relative variation of SARS-CoV-2 transmission associated with influenza. In writing the equations, we implicitly assume that the impact of influenza on SARS-CoV-2 transmission, if any, is short-lived and does not extend long after influenza infection.

Finally, we incorporated an observation model that related the dynamics of SARS-CoV-2 infection to that of COVID-19 mortality, taking into account the fact that only a fraction of infections results in death and that, among those, death occurs some time after symptom onset (Flaxman et al., 2020; Verity et al., 2020; Khalili et al., 2020). We assumed an average duration of pre-symptomatic of 2.5 days, resulting in an average incubation period of 6.5 days, in broad agreement with previous empirical studies (Khalili et al., 2020; Tindale et al., 2020). Hence, individuals in the first infected state (I1) were considered pre-symptomatic, and the onset of symptoms was assumed to coincide with the transition from I1 to I2. The onset-to-death time was then assumed to be Erlang distributed with shape parameter 5 and mean 1/κ = 17.8 days (coefficient of variation of 0.45), the value estimated in a previous epidemiological study (Verity et al., 2020). In a sensitivity analysis, we also tested a mean onset-to-death time of 1/κ = 13 days, the lower bound estimated in a meta-analysis (Khalili et al., 2020). According to previous studies in European countries, the infection fatality ratio (IFR) typically ranged from 0.5% to 1% early in the epidemic (Salje et al., 2020; O’Driscoll et al., 2021; Pastor-Barriuso et al., 2020). We fixed the IFR to μ = 0.01 in the base model, but we considered an alternative value of 0.005 in a sensitivity analysis. Given those assumptions, the observation model was modeled by the following set of ordinary differential equations: Q ˙1=2γμI1−5κQ1Q ˙i=2,…,5=5κQi−1−QiD ˙M=5κQ5

Here DM is the simulated number of daily deaths, modeled as an accumulator variable and reset to 0 at the end of each day. The observed number of daily deaths, DO, was modeled using a negative binomial distribution with mean DM and over-dispersion kD (i.e.,  VDO|DM=DM+kDDM2), a standard distribution used in previous modeling studies (Flaxman et al., 2020; King et al., 2015).

As in Flaxman et al. (2020), simulations were started 30 days before the date from which the cumulative observed death count first exceeded 10. At that date, we assumed that E1(0) individuals had been exposed to SARS-CoV-2; other individuals were assumed susceptible to infection (i.e.,  S(0) = N − E1(0)), and all other compartments were initialized to 0.

Stochastic variant and modeling of superspreading

The stochastic variant of the model was implemented as a continuous-time Markov process approximated via a multinomial modification of the τ-leap algorithm (He, Ionides & King, 2010), with a fixed time step of Δt = 10−1 day. To model the effect of superspreading events—a key feature of SARS-CoV-2 transmission dynamics (Althouse et al., 2020)—, extra-demographic stochasticity was added to the transmission rate β(t). Specifically, as proposed by Kain et al. (2020), at every time step we drew a value of β0 from a Gamma white noise distribution: β0∼ΓWNσ=R0I1+I2kΔt,μ=R0γ

with mean µand variance μσ2. Here k represents the dispersion parameter of the Negative-binomial distribution for the individual reproduction number (with mean R0 and variance R0+R02k), as estimated in previous studies (Lloyd-Smith et al., 2005; Endo et al., 2020). As in Kain et al. (2020) and in keeping with empirical estimates from contact tracing studies of SARS-CoV-2 (Endo et al., 2020), we fixed k = 0.16.

Model estimation

Following the method presented in Flaxman et al. (2020), we estimated unknown model parameters using observed COVID-19 mortality data alone. Indeed, because of the initially limited testing capacity (typically reserved to severe cases or high-risk groups), mortality data were arguably more reliable than case data early in the epidemic in most countries (Flaxman et al., 2020). By incorporating known epidemiological parameters (key among those the onset-to-death time, the IFR, and the generation time), however, the method allows back-calculating infection rates from observed death rates. Hence, in addition to the dynamic of mortality, we also reconstructed the dynamic of infection and, as a validation, compared it to external epidemiological data—like cross-sectional seroprevalence estimates—when available.

The following five parameters were estimated from the data:

1. The basic reproduction number, R0. According to a previous meta-analysis (Alimohamadi, Taghdir & Sepandi, 2020), this parameter was searched in the interval 1–10.

2. The impact of non-pharmaceutical control measures, b. The lower bound of the search interval of this parameter was fixed to 0.5, such that the maximal value of the stringency index (s = 100) corresponded to a minimal reduction of SARS-CoV-2 transmission of 50% (Flaxman et al., 2020).

3. The impact of influenza on SARS-CoV-2 transmission, βF (search interval: ℝ).

4. The initial number of individuals exposed to SARS-CoV-2, E1(0) (search interval: 0–104).

5. The over-dispersion in death reporting, kD (search interval: ℝ+).

A summary list of fixed and estimated model parameters is presented in Table 1.

All parameters were transformed to be estimated on the real line, using a log transformation for positive parameters and the extended logistic function fθ= logθ−ab−θ for parameters constrained in the interval [a, b]. The maximum iterated filtering algorithm (MIF2, Ionides et al., 2015), implemented in the R (version 3.6.3) package pomp (King, Nguyen & Ionides, 2016; R Core Team, 2020) (version 2.7), was used to estimate model parameters. The R checkpoint package was used to freeze all the packages’ version at the date of April 3, 2020 (De Vries, 2020). The estimation was completed in several steps, starting with trajectory matching to identify good starting parameters for MIF2, followed by 100 independent runs of MIF2 to locate the maximum likelihood estimate (MLE). Each MIF run had 150 iterations with 5,000 particles, geometric cooling, and a random walk standard deviation of 0.1 for the initial condition E1(0) and of 0.02 for the other parameters. The log-likelihood of every parameter set was calculated as the log of the mean likelihood of 5 replicate particle filters, each with 20,000 particles. The profile likelihood was calculated to verify the convergence of MIF2 and to derive an approximate 95% confidence interval for the parameter βF (Raue et al., 2009)—the parameter of key interest in our study. For the other parameters, a parametric bootstrap was used to calculate approximate 95% confidence intervals, by re-estimating the parameters for each of 200 synthetic datasets simulated at the MLE (Domenech de Celles et al., 2018; Domenech de Celles et al., 2019). Compared with the profile likelihood, the parametric bootstrap requires less computation and was found to perform well in previous applications (Domenech deCelles et al., 2018; Domenech de Celles et al., 2019).

Sensitivity analyses

To verify the robustness of the parameter estimates, we conducted six sensitivity analyses (further detailed in the Supplementary Results). First, we estimated an extended model in which the reduction of SARS-CoV-2 transmission was allowed to scale non-linearly with the stringency index. Second, to test the possible presence of other variables confounded with influenza, we estimated a model that included an exponential trend in the transmission rate of SARS-CoV-2. Finally, we varied the fixed value of 3 parameters and re-estimated the parameters of the base model. Specifically, we tested two alternative values of the average generation time (𝔼(Tg) = 5 days and 𝔼(Tg) = 7.5 days), one alternative value of the infection fatality ratio (μ = 0.005), and one alternative value of the average onset-to-death period (1κ=13 days).

Results

Parameter estimates

As shown in Fig. 1A, the number and the intensity of control measures against COVID-19 gradually increased from January until the nationwide lockdown in March, before a relaxation from May 2020. During this time period, the epidemic of influenza started in January 2020 and ended in March 2020, with a peak during February in each country (Fig. 1B and Fig. S1). Despite correlations between some parameters (in particular the reproduction number and the impact of control measures, see Fig. S4), all parameters were identifiable in each country (Table 2). Parameter estimates indicated that, during the period of co-circulation, influenza was associated with an average 1.8–3.4-fold (uncertainty range across countries: 1.1 to 5.0) population-level increase in SARS-CoV-2 transmission (Table 2 and Fig. S3). After controlling for the impact of influenza, our estimates of the basic reproduction number (R0) ranged from 1.2 (in Italy) to 3.4 (in Belgium). Although the increased transmission associated with influenza early during the SARS-CoV-2 epidemic explained the data significantly better (Table 2), a model without influenza led to higher R0 estimates (range 2.4–5.2, Fig. 2A), consistent with those of a previous study (Flaxman et al., 2020). Also in line with Flaxman et al. (2020), we found consistent evidence for a marked impact of non-pharmaceutical control measures (Table 2), which were associated with a decrease in SARS-CoV-2 transmission below the reproduction threshold from mid-March to June 2020 (Fig. 2A).

Table 1 List of model parameters.

Symbol	Meaning	Fixed value or estimation range	Comment/Source	
DE = 1/σ	Average latent period	4 days	Li et al. (2020)	
DI = 1/σ	Average infectious period	5 days	Fixed to have average
generation time of 6.5 days.
Sensitivity analyses: 2,7days	
Tg = DE + DI/2	Average generation time	6.5 days	Flaxman et al. (2020) and Bi et al. (2020)	
1/κ	Average onset-to-death time	17.8 days	Verity et al. (2020) and Flaxman et al. (2020)
Sensitivity analysis: 13 days	
µ	Infection-fatality ratio	0.01	Flaxman et al. (2020) and O’Driscoll et al. (2021)
Sensitivity analysis: 0.005	
N	Population size	Belgium: 11.50 M; Italy: 60.32 M; Norway: 5.37 M; Spain: 47.01M	2019 demographic data from the World Bank	
sit	Stringency index	fixed (covariate)	Fig. 1A	
Ft	Incidence of influenza (rescaled)	fixed (covariate)	Fig. 1B	
R 0	Basic reproduction number	1–10	Alimohamadi, Taghdir & Sepandi (2020)	
b	Impact of non- pharmaceutical control measures	0.5–2	Flaxman et al. (2020)	
β F	Impact of influenza on transmission	ℝ		
k	Dispersion of individual reproduction number	0.16	Kain et al. (2020) and Endo et al. (2020)	
k D	Over-dispersion in death reporting	ℝ+		
E10	Initial number exposed to SARS-CoV-2	0–104	Initial condition	

Table 2 Model parameter estimates in Belgium, Italy, Norway, and Spain.

For the proportion infected as of May 4, the numbers between parentheses represent a 95% prediction interval, based on 1,000 simulations at the maximum likelihood estimate. For the other parameters, they represent an approximate 95% confidence interval, calculated using either the profile likelihood (Raue et al., 2009) (parameter βF) or a parametric bootstrap (other parameters).

Quantity	Belgium	Italy	Norway	Spain	
Study period (year 2020)	13 Feb–28 Jun	29 Jan–28 Jun	25 Feb–28 Jun	06 Feb–28 Jun	
Log-likelihood (SE)	–384.4 (<0.1)	–649.5 (0.1)	–161.8 (<0.1)	–558.5 (0.2)	
Basic reproduction number (R0)	3.4
(2.5, 4.1)	1.2
(1.1, 1.4)	2.2
(1.0, 2.5)	1.4
(1.0, 1.9)	
Impact of control measures (b)	1.03
(0.96, 1.07)	0.53
(0.50, 0.61)	1.05
(0.53, 1.08)	0.75
(0.56, 0.86)	
Impact of influenza (βF)	0.8
(0.5, 1.3)	1.8
(1.5, 2.0)	1.0
(0.1, 2.0)	2.4
(1.7, 4.0)	
Initial number exposed to SARS-CoV-2 (E1(0))	100
(20, 200)	530
(260, 1000)	130
(100, 2800)	400
(170, 780)	
Over-dispersion in death reporting (kD)	7 × 10−4
(1,47)×10−4	0.07
(0.05, 0.09)	0.16
(0.01, 0.42)	0.08
(0.05, 0.10)	
Proportion infected, as of 4 May 2020 (%)	8.8
(3.7, 17.1)	5.4
(3.9, 7.3)	0.4
(0.2, 0.8)	6.0
(3.8, 8.6)	
Notes.

SE standard error, calculated using 5 replicate particle filters, each with 20,000 particles, at the maximum likelihood estimate

Figure 2 Dynamics of SARS-CoV-2 transmission and of COVID-19 mortality in Belgium, Italy, Norway, and Spain.

(A) time plot of the estimated effective reproductive number (Re). In each panel, the black line represents the maximum likelihood estimate and the grey ribbon the 95% confidence interval (calculated based on the likelihood profile of the influenza impact parameter, cf. Table 2) in each country. The dotted black line represents the effective reproduction number estimated from a model without influenza (i.e., with the influenza impact parameter fixed to 0 and the other parameters estimated from the data). The horizontal grey line is at Re = 1. (B) time plot of the simulated and observed numbers of daily deaths caused by SARS-CoV-2. In each panel, the light grey lines represent 1,000 model simulations at the maximum likelihood estimate, with one simulation highlighted in dark grey; the black line represents the actual death counts. In A and B, the x-axis and the y-axis values differ for each panel.

Model evaluation

Visual inspection of simulations suggested that our model correctly captured the dynamics of COVID-19 mortality in every country (Fig. 2B). A more detailed model–data comparison of summary statistics Wood (2010) confirmed that our model accurately reproduced the peak time, the peak number and the total number of deaths, and the death growth exponent (Maier & Brockmann, 2020), except in Italy and Spain where the latter statistic was systematically under-estimated (Fig. S5). Our model-based estimates of the total proportion of individuals infected with SARS-CoV-2 (as of 4 May 2020, Table 2) were also comparable with those of a previous modeling study (Flaxman et al., 2020) and of a seroprevalence study conducted in early May in Spain (approximate seroprevalence estimate of 5% (Pollán et al., 2020)). Hence, our model appeared to precisely recapitulate the epidemiology of SARS-CoV-2 morbidity and mortality over a period of ∼4 months.

Sensitivity analyses

The results of the sensitivity analyses are presented in Tables S1–S3. We found little statistical evidence that the model with non-linear scaling of the stringency index outperformed the model with simple linear scaling in any country (ΔlogL ∈ [0.0, 1.1], likelihood ratio test P-value P ∈ [0.14, 1.00], Table S1). Of note, although our estimates of the impact of influenza varied little, the additional estimated parameter resulted in higher parametric uncertainty, particularly in Norway where the approximate 95% CI embraced the null value. Similarly, the model with an unexplained exponential trend in transmission did not substantially improve model fit (ΔlogL ∈ [0.0, 3.2], likelihood ratio test P-value P ∈ [0.01, 1.00], Table S2). Despite higher parametric uncertainty caused by the estimation of the trend, our results regarding the impact of influenza remained robust, but once again the approximate confidence interval embraced the null value in Norway. In addition, we found that the parameter estimates varied little when testing alternative hypotheses about the fixed value of the average generation time, of the onset-to-death time, and of the infection fatality ratio (Table S3). Finally, because our R0 estimates were lower than previous estimates in Italy and Spain, we tested an alternative model with R0 fixed to 2.5 in each country. This model led to broadly similar conclusions, although the estimated impact of influenza was lower both in Italy (βF = 1.4, approximate 95% CI [1.2–1.5], logL =  − 692.3 [SE < 0.1]) and in Spain (βF = 0.7, approximate 95% CI [0.4–1.0], logL =  − 567.7 [SE = 0.1]). In sum, our main result about the impact of influenza remained robust to a variety of alternative assumptions regarding the epidemiological traits of SARS-CoV-2 and the modeled impact of control measures.

Discussion

The main goal of this study was to test the hypothesis that influenza impacted the epidemiological dynamics of SARS-CoV-2, building on previous experimental evidence of a positive interaction between the two viruses (Smith et al., 2020; Ziegler et al., 2020; Bai et al., 2021). To do so, we developed a semi-mechanistic, population-based model of SARS-CoV-2 transmission and of COVID-19 mortality, which simultaneously incorporated the impact of non-pharmaceutical control measures and of influenza. Using likelihood-based statistical inference techniques, we confronted this model with mortality incidence data in four European countries to systematically test a range of assumptions about the possible impact of influenza and of control measures. In keeping with previous studies (Flaxman et al., 2020), we found robust and consistent evidence that control measures markedly reduced the transmission of SARS-CoV-2. In addition, we also found consistent evidence suggesting that co-circulation of influenza transiently facilitated the transmission of SARS-CoV-2 early in the epidemic in Europe.

Our study has a number of important limitations. First, as in other studies (Flaxman et al., 2020; Li et al., 2020; Kucharski et al., 2020) and because of a lack of appropriate age-specific data (for example on the temporal changes in the contact matrix and in the incidence of influenza), our model was not age-structured, even though many aspects of COVID-19 and of influenza epidemiology—like disease severity and lethality—vary markedly with age (Verity et al., 2020). The susceptibility to SARS-CoV-2 infection was also found to increase with age (Davies et al., 2020), a finding potentially explained by lower baseline expression of the ACE2 receptor in children (Bunyavanich, Do & Vicencio, 2020). A testable prediction of our model, therefore, is that influenza should be associated with a transient increase in susceptibility to SARS-CoV-2 infection, commensurate with the variations of influenza incidence over age. Second, we modeled the impact of non-pharmaceutical control measures using a simple, linear function scaling the stringency index to the reduction of SARS-CoV-2 transmission. Even though this simple hypothesis provided a more parsimonious fit, that result may be specific to Europe, where control measures gradually increased in number and in intensity (Fig. 1A). In general, the association is likely non-linear (e.g., if a high-impact intervention like a lockdown is implemented early on), and we therefore recommend testing a variety of scaling functions. More generally, although our model builds on a previously validated method to estimate the time-varying reproduction number (Flaxman et al., 2020), we acknowledge that the stringency index may not fully capture temporal variations in SARS-CoV-2 transmission, in particular behavioral changes outside of what was mandated by governments. Even though we tested a model with an unexplained trend in transmission (Table S2), more complex temporal functions may be required to fully capture such changes. Third, we did not incorporate climate into our model, even though, as for other respiratory viruses, environmental variables like temperature and humidity may affect the transmission of SARS-CoV-2. According to previous studies conducted in a variety of locations worldwide, however, the impact of weather on the SARS-CoV-2 epidemic appears to have been modest, at least during the first wave in early 2020 (Jüni et al., 2020; Gaudart et al., 2021; Sehra et al., 2020). These findings are also consistent with epidemiological theory, which predicts that, because of lack of population immunity, the initial pandemic trajectory may be relatively insensitive to climate (Baker et al., 2020). Fourth, we did not specifically model fully asymptomatic cases, which may represent a large fraction of SARS-CoV-2 infections (Li et al., 2020). The omission of asymptomatic infections may lead to biased R0 estimates if their duration significantly differs from that of symptomatic infections (Park et al., 2020). A previous study, however, estimated that the duration of both types of infection is comparable (Li et al., 2020), such that our estimates should be robust in more complex model structures. Finally, we assessed only the impact of influenza, because of its high prevalence and period of overlap with SARS-CoV-2 in early 2020 in Europe and of the availability of high-quality data (Kramer & Shaman, 2019). Nevertheless, other respiratory viruses, like RSV and rhinoviruses (Dee et al., 2021), may also interact with SARS-CoV-2 and could be considered.

Acknowledging these limitations, our model makes at least two other predictions that could be tested to provide confirmatory evidence. First, even though our results did not allow to distinguish between higher transmissibility or higher susceptibility in individuals co-infected with influenza and SARS-CoV-2, previous experimental work suggests that the latter mechanism may operate, as a result of up-regulation of the ACE2 receptor caused by influenza infection (Smith et al., 2020; Ziegler et al., 2020). Hence, we predict that a recent influenza infection should be an independent risk factor for subsequent SARS-CoV-2 infection. Estimates of the frequency of co-detection of influenza and SARS-CoV-2 by polymerase chain reaction (PCR) testing in nasopharyngeal swabs were highly variable in previous studies (range 0–60% (Thindwa et al., 2020; Ozaras et al., 2020)). Although the marked seasonality of influenza in temperate regions likely explains in part the low frequency found in some studies (Ozaras et al., 2020), we propose that differences in the natural history of influenza and SARS-CoV-2 infections also lead to a systematic under-estimation of co-infection. Specifically, because the incubation period of SARS-CoV-2 infection (estimated to average 5.7 days (Khalili et al., 2020)) exceeds that of influenza (A, 1.4 days or B, 0.6 days (Lessler et al., 2009)), it is likely that, by the time SARS-CoV-2 infection becomes detectable, influenza no longer is. To make that statement more precise, we calculated the probability of detectability of a co-infection, with influenza first then SARS-CoV-2 (Table S4). Assuming that influenza is detectable by PCR up to 4–5 days after (Carrat et al., 2008), and SARS-CoV-2 from 2–4 days before (Tindale et al., 2020), symptom onset, we find that a large fraction (30–50%) of co-infections may not be detectable at all. These results may help explain the low frequency of co-detection found in some studies (Kim et al., 2020), and suggest that the time window of co-detectability may be too short to adequately infer the association between influenza and SARS-CoV-2 using PCR testing. Serological studies comparing the prevalence of antibodies against influenza in SARS-CoV-2 cases and non-cases may therefore be required to test the prediction that influenza is a risk factor for SARS-CoV-2 infection. Second, we predict that, all else being equal, individuals vaccinated against influenza should be at lower risk of SARS-CoV-2 infection than those unvaccinated. The findings of a negative association between influenza vaccine coverage and COVID-19 mortality in ecological studies (in Italy (Marín-Hernández, Schwartz & Nixon, 2020) and in other countries (Arokiaraj, 2020)) and of a lower risk of SARS-CoV-2 infection in influenza vaccinees in some individual-level epidemiological studies (reviewed in Riccio et al., 2020) are consistent with our prediction, but further epidemiological investigations are needed. Importantly, our results can explain these findings as the direct effect of influenza vaccines on influenza infection, instead of indirect effects on non-influenza pathogens (e.g., as a result of trained immunity) (Salem & El-Hennawy, 2020).

With the likely prospect of COVID-19 becoming endemic, the potential interactions of SARS-CoV-2 with other respiratory pathogens—in particular respiratory viruses—may become a key public health issue. In this context, our results suggest that influenza could facilitate the circulation of SARS-CoV-2 and therefore increase the burden of COVID-19. As outlined above, these results are consistent with several lines of experimental (Bai et al., 2021; Smith et al., 2020; Ziegler et al., 2020) and epidemiological (Riccio et al., 2020; Arokiaraj, 2020; Marín-Hernández, Schwartz & Nixon, 2020) evidence. We note, however, that a previous study proposed that influenza and SARS-CoV-2 have competitive interactions (Pinky & Dobrovolny, 2020). Specifically, using a within-host model of viral replication Pinky and Dobrovolny found that the low growth rate of SARS-CoV-2 may result in limited access to target cells and therefore suppression by other respiratory viruses (Pinky & Dobrovolny, 2020). Although this mechanism may be generally relevant for respiratory viruses, the ability of influenza to up-regulate ACE2 (Smith et al., 2020; Ziegler et al., 2020)—a feature not included in the within-host model (Pinky & Dobrovolny, 2020)—could counteract this mechanism and explain the increased infectivity of SARS-CoV-2 found experimentally (Bai et al., 2021). Of note, in keeping with Pinky & Dobrovolny (2020), another experimental study found evidence that the interferon response caused competitive interactions between rhinoviruses and SARS-CoV-2 (Dee et al., 2021). Hence, these different biological mechanisms suggest that every respiratory virus may interact with SARS-CoV-2 in a highly specific way, with influenza being unique in its ability to up-regulate ACE2 and to increase SARS-CoV-2 infectivity (Bai et al., 2021).

Conclusions

In conclusion, our results suggest that influenza virus infection could have increased the transmission of SARS-CoV-2 and facilitated its spread during the early 2020 epidemic of COVID-19 in Europe. Hence, an increase in the uptake of influenza vaccines may be called for, not only to reduce hospitalizations due to influenza infections (Ozaras et al., 2020; Paget et al., 2020), but also to reduce their downstream impact on SARS-CoV-2 transmission and on COVID-19 mortality. More generally, taking into account the microbial environment of SARS-CoV-2 may be essential, not only to better understand its epidemiology, but also to enhance current and future infection control strategies.

Supplemental Information

Supplemental Information 1 Supplementary Figures and Tables

Click here for additional data file.

We thank Arturo Zychlinsky and Klaus Osterrieder for helpful comments on the manuscript. Computations underlying the present analysis were performed at the Max Planck Computing and Data Facility (MPCDF).

Additional Information and Declarations

Competing Interests

Author Contributions

Data Availability

Matthieu Domenech de Cellès received postdoctoral funding (2017–2019) from Pfizer and consulting fees from GSK. Jean-Sebastien Casalegno declares no competing interests. Bruno Lina is chair of the ISC for the Global Influenza Surveillance Network, and co-chair of the Global Influenza and RSV initiative; he is also a member of the French COVID- 19 Scientific Committee (no personal income for all these activities). Lulla Opatowski has received consulting fees from WHO for work on antimicrobial resistance and funding from Pfizer (2017–2019).

Matthieu Domenech de Cellès conceived and designed the experiments, performed the experiments, analyzed the data, prepared figures and/or tables, authored or reviewed drafts of the paper, and approved the final draft.

Jean-Sebastien Casalegno, Bruno Lina and Lulla Opatowski conceived and designed the experiments, authored or reviewed drafts of the paper, and approved the final draft.

The following information was supplied regarding data availability:

The data used for this analysis are freely available from the databases cited in the references and have been compiled into a database available from Edmond, the Open Data Repository of the Max Planck Society: https://dx.doi.org/10.17617/3.7r. All R programming codes are also available from this repository.

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

Ziegler et al. (2020) Ziegler CGK Allon SJ Nyquist SK Mbano IM Miao VN Tzouanas CN Cao Y Yousif AS Bals J Hauser BM Feldman J Muus C Wadsworth n Marc H Kazer SW Hughes TK Doran B Gatter GJ Vukovic M Taliaferro F Mead BE Guo Z Wang JP Gras D Plaisant M Ansari M Angelidis I Adler H Sucre JMS Taylor CJ Lin B Waghray A Mitsialis V Dwyer DF Buchheit KM Boyce JA Barrett NA Laidlaw TM Carroll SL Colonna L Tkachev V Peterson CW Yu A Zheng HB Gideon HP Winchell CG Lin PL Bingle CD Snapper SB Kropski JA Theis FJ Schiller HB Zaragosi L-E Barbry P Leslie A Kiem H-P Flynn JL Fortune SM Berger B Finberg RW Kean LS Garber M Schmidt AG Lingwood D Shalek AK Ordovas-Montanes J HCA Lung Biological Network Electronic address: lung-network@humancellatlas.org, and HCA Lung Biological Network 2020 SARS-CoV-2 receptor ACE2 is an interferon-stimulated gene in human airway epithelial cells and is detected in specific cell subsets across tissues Cell 181 1016 1035 10.1016/j.cell.2020.04.035 32413319 PMC7252096