# Peer review of "Estimating the impact of influenza on the epidemiological dynamics of SARS-CoV-2"

_PeerJ, doi:10.7717/peerj.12566_

## Round 0.1 · original submission · Major Revisions

Both reviewers expressed substantial concerns over technical aspects of your manuscript. Please address all the comments including (1) the appropriateness to justify the quantitative feature of the model by fitting it to mortality data alone (e.g. any additional data support?) and (2) the rationale of enhancement of SARS-CoV2 transmission by influenza (in relation to existing evidence of rhinovirus).

·

Basic reporting

Authors showed the possible impacts of seasonal influenza on the SARS-CoV-2 transmissions, by fitting the compartment model with the number of COVID-19 deaths in European countries. The manuscript is well written and provides insight into interrelationship between the transmission dynamics of influenza and COVID-19. However, I have some major concerns in terms of the model design and evaluation. I hope this suggestion make this manuscript to have the highest impact.

Experimental design

1. I am not convinced that the actual transmission dynamics of COVID-19 can be reconstructed by fitting the age-aggregated model with the number of deaths. As authors mentioned in the Limitation part, the infection fatality risk of COVID-19 is suggested to considerably increase by age, moreover, young adults contracted with COVID-19 are likely to be a mild case (but they are infectious). Thus, considering this crucial characteristic of COVID-19, without using the age-structured model, I believe reconstructing the actual dynamics by fitting with the number of deaths might be highly limited.

2. I would be inclined to mention why authors assumed the negative binomial distribution in death reporting. I think the most crucial advantage of death data is under-ascertainment issue can be partially removed, since the COVID-19 death cases are highly likely to be observed. Did the countries in this study have huge difficulties in death reporting? or were there any circumstance with huge under-reporting in the number of deaths? It would be great if authors can clarify the reason of introducing the negative binomial distribution in death reporting.

Validity of the findings

1. The model evaluation was conducted by comparing the number of observed and modeled deaths. However, authors did not show that the model integrated with the influenza dynamics can provide better insight into actual COVID-19 dynamics, compared to the conventional COVID-19 model. The fitting results looks plausible and authors also show some published papers about the possible impacts of influenza. However, I think it is not enough to say the model incorporated with the influenza can provide a better performance and can be used as an evidence. I suggest authors to compare the AIC value between the proposed and conventional model to show the performance of model.

Additional comments

[Minor comments]
1. L142: I would like to suggest to change the abbreviation of the stringency index to other word. Since s(t) seems like the number of susceptible at time t (in the compartment model), it can make readers confused.
2. Could authors show the estimated overdispersion parameter in death reporting (k_D) in the Table 2?

Reviewer 2 ·

Basic reporting

The manuscript meets all basic reporting standards.

Experimental design

While I find the results of the study intriguing, I think there are some further assessments needed to make the results more conclusive.

1. Similar to the study showing suppression of SARS-CoV-2 infections by rhinovirus (Dee et al. cited in the current manuscript), there is a study indicating that SARS-CoV-2 infections could be suppressed by influenza (Pinky and Dobrovolny, (2020) J. Med. Virol.). How does this fit into the assumption that SARS-CoV-2 transmission is enhanced by influenza?

2. The authors mention that they fit the model with beta_f = 0 to the data, but they do not present these results in the manuscript. I think the parameter estimates and log likelihoods for this model need to be included in the manuscript (or supplementary material with the other models). After all, if the log likelihood for a model without the effect of influenza is similar to the log likelihood of the model with influenza, then the case for the effect of influenza is pretty weak.

3. I think the authors also need to include correlation plots of the estimated parameters. It could be that the assumed effect of influenza can be compensated for by altering other model parameters.

4. The set range for some of the estimated parameters might be too small. The 95% CI for b and R0 are hitting the lower boundary for some of the data sets.

5. Why are two different methods used to estimate 95% CI for the estimated parameters (Table 2 caption)?

6. This wasn't entirely clear to me, but was the model fit only to daily deaths? Or were case counts also included?

7. The discussion around lines 345-354 is actually presenting a new result and should probably be included in the results section.

8. The authors should perform additional sensitivity analysis on the number of stages assumed in each of the transitions. As n --> infinity, the number of stages has little effect on the estimated parameters, but the authors have fixed each transition to 2 stages, so even a change to 3 stages might change the estimated parameter values.

Validity of the findings

The conclusions are well stated, although as outlined above, I think more evidence is needed to support their claim.

---

## Round 0.2 · Major Revisions

One of the reviewers strongly indicated the technical flaw of using age-independent IFR. I agree. I'm not sure if the authors come up with anything other than likelihood-based inference, but the IFR part must be addressed.

·

Basic reporting

Authors revised the manuscript and overall the revised manuscript is well written in accordance to comments. I appreciate authors' effort on it. However, I am not still convinced that the validity of proposed model is sufficiently secured to assess the impact of seasonal influenza on the transmission dynamics of COVID-19.

Experimental design

Unlike the early phase of epidemic, detailed information on the characteristics of SARS-CoV-2 transmission has been identified. Thus, at this stage, I do not fully agree that reconstructing the transmission dynamics of COVID-19 by fitting the age-aggregated model with daily deaths using an identical infection fatality risk across all age is sufficient, especially, to explore the possible association with the seasonal influenza. In addition, although the estimated confidence interval of proportion of infected individuals among all population includes the estimated value from sereprevalence data (Table 2), considering an absolute number of the 1% of total population in each countries, I think the estimates with the broad confidence intervals is not so informative to show the validity of the reconstructed model.

Validity of the findings

I fully understood that the present study relies on the likelihood-based inference, however, considering the different number of used parameters, I do not think that only showing the likelihood can clearly clarify that the incorporation of seasonal influenza can significantly improve the model. Despite of my opinion, the shown table shows the model incorporated with seasonal influenza shows better performances, although I still have some worries on the validity of model as mentioned in the upper comment.

Reviewer 2 ·

Basic reporting

No comment

Experimental design

The authors have incorporated many of the additional checks, strengthening the support for their findings.

Validity of the findings

No comment

Additional comments

The authors have addressed my concerns to my satisfaction.

---

## Round 0.3 · accepted · Accept

I have read the paper and the comments of the authors/reviewers, and in my opinion this paper should be accepted in its current state. Everything has been correctly explained and the limitations about the inclusion of other factors, like age, has been detailed to open new lines of research.

Therefore, I am pleased to indicate the acceptance of your work for publication in PeerJ.

Congratulations!